# Recent Advances in Surface-Enhanced Raman Scattering Magnetic Plasmonic Particles for Bioapplications

**DOI:** 10.3390/nano11051215

**Published:** 2021-05-04

**Authors:** Kim-Hung Huynh, Eunil Hahm, Mi Suk Noh, Jong-Hwan Lee, Xuan-Hung Pham, Sang Hun Lee, Jaehi Kim, Won-Yeop Rho, Hyejin Chang, Dong Min Kim, Ahruem Baek, Dong-Eun Kim, Dae Hong Jeong, Seung-min Park, Bong-Hyun Jun

**Affiliations:** 1Department of Bioscience and Biotechnology, Konkuk University,120 Neungdong-ro, Gwangjin-Gu, Seoul 05029, Korea; huynhkimhung82@gmail.com (K.-H.H.); greenice@konkuk.ac.kr (E.H.); phamricky@gmail.com (X.-H.P.); susia45@gmail.com (J.K.); pie3000@naver.com (D.M.K.); wisdomfairy@konkuk.ac.kr (A.B.); kimde@konkuk.ac.kr (D.-E.K.); 2Medical Device & Bio-research Team, Bio-medical & Environ-chemical Division, Korea Testing Certification, Gunpo, Gyeonggi-do 15809, Korea; pourlady@ktc.re.kr; 3Center for Convergent Research of Emerging Virus Infection, Korea Research Institute of Chemical Technology, Daejeon 34114, Korea; jonghwan@krict.re.kr; 4Department of Chemical and Biological Engineering, Hanbat National University, 125 Dongseo-daero, Yuseong-gu, Daejeon 34158, Korea; sanghunlee@hanbat.ac.kr; 5School of International Engineering and Science, Jeonbuk National University, 567 Baekje-daero, Deokjin-gu, Jeonju-si, Jeollabuk-do 54896, Korea; rho7272@jbnu.ac.kr; 6Division of Science Education, Kangwon National University, 1 Gangwondaehakgil, Chuncheon-si, Gangwon-do 24341, Korea; hjchang@kangwon.ac.kr; 7Department of Chemistry Education, Seoul National University, 1 Gwanak-ro, Gwanak-gu, Seoul 08826, Korea; jeongdh@snu.ac.kr; 8Center for Educational Research, Seoul National University, 1 Gwanak-ro, Gwanak-gu, Seoul 08826, Korea; 9Department of Urology, Department of Radiology, Molecular Imaging Program at Stanford, Stanford University School of Medicine, Stanford, CA 94305, USA

**Keywords:** Surface-Enhanced Raman Scattering (SERS), magnetic nanoparticles, plasmonic nanoparticles, detection, drug delivery, cancer therapy, biological application

## Abstract

The surface-enhanced Raman scattering (SERS) technique, that uses magnetic plasmonic particles (MPPs), is an advanced SERS detection platform owing to the synergetic effects of the particles’ magnetic and plasmonic properties. As well as being an ultrasensitive and reliable SERS material, MPPs perform various functions, such as aiding in separation, drug delivery, and acting as a therapeutic material. This literature discusses the structure and multifunctionality of MPPs, which has enabled the novel application of MPPs to various biological fields.

## 1. Introduction

In 1974, Fleischmann et al. first reported surface-enhanced Raman scattering (SERS) when they observed an unexpectedly strong Raman signal from pyridine adsorbed on a roughened silver electrode [1]. Since then, many studies have verified that the SERS phenomenon is related to electromagnetic and chemical effects [1,2]. Currently, SERS is a popular spectroscopic technique based on the plasmon-assisted scattering of molecules on or near metal nanostructures. It is widely used owing to several advantages, such as simple sample preparation, capability of detecting multiple analytes, and non-destructive analysis at a high sensitivity level [3,4,5,6,7,8,9,10,11].

Magnetic nanoparticles (MNPs) are 1–100 nm sized nanoparticles that can be manipulated using external magnetic fields [12,13]. Owing to their excellent physicochemical properties, studies on MNPs as optical probes or sensors have been undertaken in a wide range of fields, such as biotechnology/biomedicine [12,13,14,15,16]. However, bare MNPs possess high chemical activity; thus, they easily agglomerate or oxidize [12]. Therefore, surface functionalization of MNPs is required in many applications. Combining MNPs with other materials, such as plasmonic nanoparticles to create magnetic plasmonic particles (MPPs), can also help overcome these shortcomings.

Integrating MNPs into SERS has multiple benefits owing to the combination of plasmonic and magnetic properties, including high sensitivity, finger-print specificity, non-destructive detection of SERS characteristics, rapid separation, and simple external data gathering. Additionally, time-consuming complex matrix extraction of analytes is unnecessary [13,17,18]. This study reviews existing information on the structure of MPPs, their functions in magnetic SERS, and the biological applications of magnetic SERS materials.

## 2. Types of MPPs

Magnetic SERS requires materials possessing magnetic and plasmonic properties. Several kinds of MPPs have been designed and modified through the addition of functional molecules as per the requirement. This section explores the features and properties of a variety of MPPs.

In core-shell type MPPs, the magnetic and plasmonic parts are combined into one particle, typically having a magnetic core with a plasmonic shell structure. These are also known as combined-type MPPs. This type acts as a multifunctional material with both magnetic and plasmonic characteristics, such as moving under a magnetic field, and thus is a material potentially suitable for use in SERS. In contrast, separate-type MPPs have the magnetic and plasmonic parts in two distinct particles. The advantage of creating dual particles is that the magnetic particles, which can be manipulated by external magnetic fields, and the plasmonic particles facilitate SERS measurement.

Nanoparticles (NPs) can be labeled by surface modification with molecules like Raman label compounds or remain as label-free NPs. Generally, label-free NPs are used in their natural form without modification for direct detection of target molecules. The intensity of the intrinsic SERS spectrum depends on the target affinity of the NP surfaces and the target concentration [18,19]. Although label-free NPs can separate and detect targets using the intrinsic and specific SERS spectrum, an impure target can cause non-specific signals, which leads to difficulty in identification. Therefore, to ensure specific detection, NPs are modified with labeling molecules and bio-ligands—such as antibodies [18,20,21,22,23,24], aptamers [25,26,27], and polysaccharide (chitosan [28])—that bind with specific targets. Labeled MPPs can only recognize the target, and thus the detection accuracy increases.

MPPs are also classified according to their chemical composition: monometallic, alloyed, and assembly particles. The magnetic and plasmonic parts of monometallic MPPs are facilitated by a single type of metal, such as Fe_3_O_4_, Ag, or Au. This is a common type of MPPs [17,18,20]. Alloyed MPPs owe their magnetic and plasmonic parts to two or more metals, such as Co-Fe_2_O_4_ [29,30], Mn-Fe_3_O_4_ [25], Fe_3_O_4_-TiO_2_ [31], Fe_2_Ni [32], and Au-Ag NPs [22,25]. Their characteristics are strongly dependent on the structure and composition of the nanomaterials [33]. Alloy metal NPs are usually employed to obtain synergistic effects, such as optical tuning and enhanced stability, owing to their hybrid characteristics.

Assembly MPPs are MPPs combined with non-metals, such as polyethylenimine (PEI) [34,35,36,37], SiO_2_ [38,39,40], graphene oxide [32,41,42,43], and poly(N-isopropylacrylamide) (pNIPAM) [44]. PEI is a hydrophilic macromolecule from the primary amine group; therefore, it is introduced as an interlayer to significantly improve the dispersion and adsorption of negatively charged Au NPs [36,37]. When looking to identify bacteria in solution, PEI is introduced as a positively charged outer layer to promote strong electrostatic interaction with negatively charged bacterial pathogens [34]. In SERS, a silica shell has proven popular due to its advantageous properties, such as chemical inertness, transparency, hydrophilicity, and biocompatibility [45]. Silica encapsulation ensures the particles are well-dispersed in a solution, have long-term stability, and are suitable for surface modification via silane-coupling chemistry, thereby ensuring bioapplication [46,47]. Additionally, graphene and graphene derived materials, such as graphene oxide and reduced graphene oxide, are also promising substrates for SERS techniques [42]. These substrates are a monolayer of carbon atoms that undergo π–π stacking or electrostatic bonding with aromatic compounds. This greatly enhances the substrates’ ability to adsorb aromatic molecules [42,43,48]. Additionally, graphene has other benefits when used with metals in a hybridized form. A combination of graphene with noble metal nanostructures exhibits higher SERS enhancement [42], and graphene with magnetic metal has good electrical conductivity and stability when biomolecules immobilize onto surfaces [49]. Finally, graphene can reduce SERS noise.

Recently, MPPs have been designed with increasingly specific functions. For example, core-shell type MPPs with antibody labeling, have been fabricated from an antibody immobilized popcorn shaped magnetic core with gold shell particles. These MPPs have been used as a multifunctional nanomaterial in magnetic separation, SERS imaging, and photothermal destruction of bacteria [24]. The MPPs are made in a two-step process, as shown in Figure 1. First, Fe NPs are synthesized by mixing a metal precursor, stabilizer, and reducing agent—iron chloride (FeCl_3_), tri-sodium citrate (TSC), and sodium borohydride (NaBH_4_), respectively. Second, hydrogen tetrachloroaurate (HAuCl_4_⋅3 H_2_O) and CTAB act as a shape-templating surfactant to form the gold shape. When used, the amine group surface modifications on the MPPs conjugate with antibodies.

In another example, Fe and Ag MPPs clusters [50] (Figure 2) are a combination of magnetic Fe_3_O_4_ NPs and plasmonic Ag NPs, and have been utilized for atrazine detection in pesticide polluted water. Through a solvothermal reaction, Ag NPs are formed by the decomposition of silver oleate. Then, smaller Fe_3_O_4_ NPs slowly diffuse and surround the surfaces of the Ag NPs, hence forming MPP clusters. Atrazine can be isolated, concentrated by the magnetic properties, and then detected by SERS. In a third example, a dual-particle style MPP, with labeling, is shown in Figure 3. These MPPs are formed when monodispersed silver-coated magnetic NPs are synthesized and conjugated with aptamer(1). The gold core and silver shell plasmonic nanoparticles, labeled with a SERS tag, 5,5′-dithiobis(2-nitrobenzoic acid); (DTNB), then conjugate with aptamer(2). The MPPs are applied for the specific identification of Staphylococcus aureus bacteria using SERS [25]. The bacteria are bonded by the magnetic NPs and separated by a magnetic bar, where SERS can then confirm the presence of the DTNB tag.

## 3. Bioapplication of MPPs

Among various nanoparticles, plasmonic NPs and magnetic NPs are widely used in biological applications, but each have certain drawbacks that can be complimented by the other type of the NPs [51]. For example, the weakness of magnetic NPs such as toxicity [52,53], aggregation [54,55,56], easy corrosion in water [52] makes it difficult to work well in biological environments [52,53,54,55,56], while Au NPs hardly detect small amounts (giving low signal) of molecules in the mixture by SERS technology [57,58]. Once combined (e.g., in iron gold core- shell MPPs), the Au shell helps increase the biocompatibility of the hybrid, and the magnetic core helps increase the sensitivity in SERS detection of the hybrid. MPPs have characteristics of both magnetic and plasmonic metals. Therefore, they can be used as both SERS substrates and traditional magnetic NPs. Also, the interplay between the magnetic core and the plasmonic shell in the hybrids exhibits a synergistic effect in thermotherapy and bioimaging [21,51]. Some studies have reported on Fe-Au core shell MPPs, where the Au shell reduces the overall magnetic property, but this property is strong enough for MPP to act as a magnetic NP [21,35,51]. For example, in Han et al.’s study, the saturation magnetization of Fe_3_O_4_ NPs is 64.9 emu g^−1^; after the Au shell is coated, the saturation magnetization decreased to 59.2 emu g^−1^ and it still retains same coercivity (30 Oe) [21]; Zhou et al. showed the saturation magnetization after coating the Au shell of Fe_3_O_4_ decreased from 77.3 emu g^−1^ to 36.8 emu g^−1^ and 17.5 emu g^−1^ but maintained the same coercivity (306 Oe) [35]; both Fe_3_O_4_-Au core shell MPPs still retain magnetic property. On the other hand, the Au shell supplements better heating in magnetic hyperthermia [21] and Fe-Au core-shell MPPs exhibited a higher transverse relaxivity comparing with Fe_3_O_4_ NPs in bioimaging [51]. Another combination between magnetic and plasmonic is Ag and Fe based on MPPs that are similar with Au-Fe MPPs in sensitive specific detection [36,38,43,59], better bioimage [28] and moreover, they show antibacterial synergistic ability [60,61]. Padilla-Cruz et al. reported that core-shell Ag-Fe spherical particles showed magnetic and antimicrobial properties [60]. These Ag-Fe NPs possessed the antibacterial synergistic effect, compared to Ag NPs or Fe NPs, against both Gram-positive and Gram-negative multidrug-resistant bacteria and yeast. The Fe NPs have not showed the antibacterial property. Values of minimal inhibitory concentration of Ag NPs and Ag-Fe NPs are the same at 125 ppm in *S.aureus* ATCC 6538 case, 62.5 and 31.25 ppm in *P. aeruginosa* ATCC 27,853 case, 31.25 and 15.62 ppm in multidrug-resistant *P. aeruginosa* case, the same at 125 ppm in *E. Coli* ATCC 11,229 case, and 125 and 62.5 ppm in *C. albicans* case [60].

MPPs are applied in a wide range of settings. Many studies have reported MPPs with multiple functions in: targeted magnetic separation and enrichment, SERS imaging, and the photothermal destruction of bacteria [24]; cancer cell targeting, separation, and imaging [51,57]; free prostate specific antigen detection, MRI, and magnetic thermotherapy [21]. It further provides enormous advantages such as low cost [18,42,50,62], high sensitivity [18,27,42,62,63], good selectivity [18,27,42,62,63] and on-line monitor [50,51], and reproducibility [18,36]. For example, Wang et al. reported about using MPPs for SERS-based bacteria sensing, which has proved that MPPs are a low-cost material for bacterial detection with specific, sensitive advantages with the limitation of detection (LOD) for *Staphylococcus aureus* (*S. Aureus*) being 10 cells mL^−1^ [18]. In comparison with other methods like ELISA based on monodisperse magnetic particles (10^4^–10^5^ cells mL^−1^ LOD [64]), PCR method (10 cells mL^−1^ LOD [65]), real-time potentiometric biosensors based on carbon nanotubes and aptamers, (8 × 10^2^ cells mL^−1^ LOD [66]), the sensitivity of MPPs based on SERS is better or equal but it is simpler in procedure while other approaches require special expensive instrumentation, trained technicians, or complicated pretreatment protocol [18]. Even in comparison with Au NP-based SERS (10–13 cells mL^−1^ LOD [67,68]), the method needs a separation step of target, MPP-based SERS only need 10 s for separation based on magnetic property [18]. For aromatic dye detection, SERS is a powerful method in this field; however, many aromatic dyes have a poor affinity in interactions with SERS material that leads to the necessity of modification of SERS substrate that can detect aromatic dyes as low as 1 nM LOD [69,70,71]. While this modification is often complex and high cost, one application of graphene oxide-wrapped MPPs as SERS substrate for aromatic dye detection exhibits the same sensitive specific detection as 1 nM LOD, and it is a lower cost material [42]. MPPs also show greater effectiveness than other approaches in protein detection. In platelet-derived growth factor BB detection, whereas MPPs based on SERS detect this protein as low as 0.1 pM LOD [62], other methods LODs are higher or similar, such as: 4 nM of Au NPs colorimetric sensor [72]; 80 pM of aptamer-based immunomagnetic electrochemiluminescence assay [73]; 68 pM of fluorescence [74]; and 0.5 pM of Au NPs based on SERS [75]. MPPs based on SERS has one more advantage which is quicker separation [62].

### 3.1. Detection and Separation

As SERS substrates, MPPs can detect targets with high sensitivity [22,76,77,78,79]. Additionally, as magnetic NPs, MPPs can be manipulated by an external magnetic field so that MPPs can be used to separate and enrich the target before SERS signal detection. Therefore, MPPs are often used to enhance the separation and detection of low concentration targets. These MPPs can also help isolate nucleic acids, proteins, and other small molecules in a complex biological matrix [59,80,81].

#### 3.1.1. Nucleic Acids

Nucleic acid diagnostics, with its high sensitivity and specificity, plays a vital role in many fields, such as biology, medicine, and environmental science. Nucleic acid fragments and nucleotides can be targeted by magnetic SERS based detection. SERS enables high sensitivity and specificity, demonstrated by its ability to separate and collect targets with concentrations as low as fM [81,82,83], such as: typical Fe_3_O_4_ and Au based MPPs for adenine detection with detection limit as 0.7 µM [84]; dual MPPs including ultrabright Au- Ag core- gap-shell NPs with Raman reporter in the gap and magnetic NPs that were utilized for DNA detection with a detection limit as low as 100 aM [85]; and Fe_3_O_4_@Ag MPPs and duplex-specific nuclease signal amplification in microRNA detection with detection limit Nucleic acid diagnostics, with its high sensitivity and specificity.

Alula et al. developed MPPs containing iron and gold NPs for adenine detection [84]. A magnetic core is produced by coprecipitation and then coated by a polymer. Au NPs are then produced on the polymer-coated surface by a photochemical reduction method. The MPPs are stable in aqueous solution, disperse well in solution, and easily attach to the target adenine. Owing to their magnetic properties, the MPPs can concentrate the target to facilitate SERS imaging. Thereby, adenine can be detected at concentrations as low as 0.7 µM [84].

Ngo et al. reported an MPPs method to discriminate wild and mutant type malaria, via DNA detection and single nucleotide polymorphism [85]. Dual MPPs with magnetic NPs and Au/Ag NPs were labeled by DNA probes matching the target sequence. A hybridized sandwich of magnetic NPs-target sequence-Au/Ag NPs is formed via specific recognition of complementary DNAs, which are concentrated at the SERS detector. The limit of detection of the detection platform was approximately 100 aM.

MicroRNAs are short single stranded RNA molecules that are the primary regulatory factor in various biological pathways, such as mRNA degradation, translational inhibition, cell proliferation, differentiation, and apoptosis. They can also cause cancer [81,86,87,88,89,90,91]. Therefore, microRNAs are used as biomarkers for several diseases, including cancer. Pang et al. fabricated a Fe_3_O_4_@Ag MPPs designed for microRNA detection [81]. The Fe_3_O_4_@Ag MPPs are modified by Raman tags-DNA probes. Target microRNA are captured on the surface of Fe_3_O_4_@Ag MPPs through DNA/RNA hybridization. The SERS signal is significantly amplified by duplex specific nuclease, yielding a detection limit of 0.3 fM. The Fe_3_O_4_@Ag MPPs have been successfully applied in microRNAs’ capture, concentration, and direct quantification, overcoming disadvantages in previous microRNAs’ diagnostic tools, such as single sequence and low abundance [81].

In addition, MPPs have been employed in determining anti-cancer drug interaction with DNA [32], and for DNA methyltransferase activity [92] measurements, which uses SERS to detect DNA fragments via interaction between the fragments and plasmonic metals. These studies showed the MPPs also supported functionalization and recovery for recycling [32], and had excellent separation ability [92].

#### 3.1.2. Protein

Like nucleic acids, proteins are key biological molecules that have a multitude of roles in organisms. There are numerous examples of proteins dispersed through an organism, including hormones, antibodies, cell-surfaced molecules, enzymes, and structural proteins. As abnormal protein quantities can lead to disease, sensitive and specific diagnosis techniques are necessary. Therefore, magnetic SERS has been applied in disease-based protein detection. Target proteins include human immunoglobulin G [20,36], prostate specific antigen [26,35,93], urinary erythropoietin [94], sepsis-specific biomarkers [95], carcinoembryonic antigen [96], thrombin, platelet derived growth factor BB, immunoglobulin E [62], and glycated hemoglobin [97]. These MPPs are designed with very specific components, such as antibodies, aptamers or chemical molecules [62,95,97]. The MPPs enable easy extraction, isolation, collection, and purification of the specific target from biological samples [62,93,94,95,96,97] and facilitate measurement at low concentrations.

#### 3.1.3. Small Molecules

As an ultrasensitive sensor, SERS not only detects macromolecules like nucleic acids and proteins, but also single, small molecules [22]. Therefore, magnetic SERS have been applied in single molecule level diagnosis. The surfaces of the MPPs are labeled with a specific target antibody [30,98,99], aptamer [27], or chemical molecule [63]. The limit of target detection concentration can be as low as fg. mL^−1^ [30]. The magnetic parts act as a high-speed extractor [98] and simple isolator [99]. Additionally, the magnetic parts concentrate [63] and aggregate particles to generate hot spots, which increase the SERS intensity and thus aid sensitivity [100]. The process is used in a wide selection of applications, such as: determination of microcystin-LR in blood plasma [98]; anthrax biomarker poly-γ-D-glutamic acid in serum [99]; cotinine and benzoylecg onine in saliva [100]; thiocyanate and tetracycline in food products [27,41]; nitrite ions in environmental, biological, and food samples [63]; and N-terminal pro-brain natriuretic peptide as a biomarker for heart failure [30].

#### 3.1.4. Cancer Diagnostic

Cancer is a group of diseases caused by abnormal growth of cells [101]. It can occur anywhere in the human body and often spreads. From the early stages of cancer, biomarkers are released [102]. If these biomarkers can be detected, then early diagnosis is possible, which aids in effective treatment. Magnetic SERS can accurately detect very low concentrations of biomarkers. In these MPPs, the plasmonic property targets the biomarkers and the magnetic property contributes to the agglomeration and the amplification of the signal. Many studies have demonstrated magnetic SERS’ usefulness in cancer marker detection with high sensitivity. MPPs are synthesized with specific biological components, including antibodies or aptamers, in order to target cancer markers or cancer cells, such as carcinoembryonic antigens [101], circulating tumor cells [102], circulating tumor DNAs [ref], RNA extract from cancer cells [81], breast-cancer cells, floating leukemia cells [57], and Bronchioalveolar stem cells [103]. Even at extremely low concentrations (~fM), the MPPs can separate and collect the target using an external magnetic field. The SERS signal is also amplified by the magnetic gathering [81]. Figure 4 shows an example of SERS detection of a cancer biomarker. In this case, the MPPs are formed from magnetic NiFe@Au and plasmonic Au NPs, with Raman labeling and suitable antibodies. The biomarker/MPPs combinations are separated, gathered by an external magnet, and detected by SERS amplification.

#### 3.1.5. Detection of Pathogens

The magnetic SERS technique is one method for the diagnosis of infectious diseases caused by pathogens, such as bacteria or viruses [25,34,104,105]. Compared with other methods—including polymerase chain reaction, enzyme-linked immunosorbent assay, and culture and colony counting of bacteria—magnetic SERS has advantages, such as simpler sample preparation requirements, improved sensitivity, and rapid and multiplexed detection [104]. Additionally, magnetic properties assist in the concentration of the target to improve separation [15], accuracy, and sensitivity [106]. Typically, pathogens can be determined directly [34,106], or indirectly via specific capture with antibodies [24,104,107,108,109] or aptamers [25,110]. The MPPs used can be magnetic plasmonic core-shell mono particles [24] or dual particles including magnetic and plasmonic particles [25]. Additionally, the MPPs can be immobilized on a surface by specific capture [107], or without labeling [34]. Once these MPPs interact with targets, the targets are gathered by the magnetic force. The pathogens have a distinct, identifiable SER signal. Magnetic SERS techniques have ultrasensitive capabilities, down to 10 pathogen cells in 1 mL [25].

### 3.2. Drug Delivery and Therapy

The needs of drug delivery and therapy often coexist, in order to carry a drug to the affected location and provide medical treatment (Figure 5). Magnetic NPs have the potential to be next-generation drug carriers because of their unique physical-chemical properties [53,56]. Magnetic NPs can move easily and rapidly to the target position under a magnetic field. The toxicity and biocompatibility of magnetic NPs can be adjusted by surface modifications, such as coatings of polyethyleneimine (PEI)-g-polyethylene glycol (PEG), chitosan, dextran, N-(2-hydroxypropyl)methacrylamide (HPMA), polymeric micelles, starches, proteins, and polyvinyl alcohol (PVA) [52,53,55]. After being immobilized with the targeting ligands (enzymes, peptides, antibodies, aptamers etc.), the magnetic NPs accurately reach the target position and release drug molecules [54,55]. For cancer and tumor treatment, magnetic NPs have been utilized as a beneficial mechanism for thermotherapy [111,112,113,114,115]. In thermotherapy (high temperature treatment), magnetic particles act as heat transferrers, causing apoptosis of the tumor cells. In a study on mice by Xie et al., effective thermotherapy was achieved with four injections, each of dose 28 mg Fe. kg^−1^ body weight, using an alternating current magnetic field of 390 kHz and 12 A, with 30 min exposure at 43 °C [114]. Additionally, plasmonic NPs, such as Au and Ag, have been investigated for application in smart drug delivery and cancer therapy [54,55,111,116,117,118,119]. Several studies have also demonstrated that Au and Ag NPs possess antitumor properties and have the potential to enhance cancer therapy [120,121,122,123,124,125,126,127,128]. Briefly, endocytosis particles are carried by vesicles to a cell’s cytoplasm and nucleus; there they produce a toxic effect and cause apoptosis or programmed cell death through reactive oxygen species, tumor necrosis factor, or interleukin-6 [117]. Meyers et al. [129] reported that peptide-targeted Au NPs successfully carried and delivered Phthalocyanine 4, an photodynamic agent, to cancer cells and then killed the cancer cells (concentration 1 µM). Ag NPs and their composite with other metals have also been investigated for antibacterial characteristics [127,130,131,132,133,134,135]. Ag NPs have been shown to effectively inhibit various pathogenic bacteria, fungi and viruses, including *Staphylococcus aureus*, *Escherichia coli*, *Pseudomonas aeruginosa,* dermatophyte, and HIV-1 [127]. The antibacterial characteristics depend on the size, shape, concentration, and charge of the particles [111,127]. The optimum parameters for antibacterial Ag NPs were found to be spherical and less than 10 nm in size, rather than triangular, linear, or cubic in shape, and larger in size [111,127]. Additionally, accumulation over time and positive surface charge can help the Ag NPs increase their antibacterial effect [127]. An Ag-Co-Cr alloy has been investigated as an antibacterial medical implant [135].

MPPs that combine magnetic and plasmonic nanoparticles have shown the same properties as the constituent magnetic NPs and plasmonic NPs. In a study by Tomitaka, magnetic plasmonic core-shell nanostar NPs demonstrated image-guided drug delivery and NIR-triggered drug release applications [51]. Under an alternating magnetic field, MPPs act as imaging contrast agents, assisting in the production of clear magnetic particle images. The Au parts of the MPPs bind the drug, which is then released at the desired location under NIR stimulation. Another study by Lal et al. demonstrated that iron oxide gold nanobowls can be guided magnetically, and that they possess distinct SERS capabilities. These characteristics show MPPs’ potential as a therapeutic drug storing and imaging contrast material [136]. Moreover, iron gold alloy NPs, in a study by Li et al., demonstrated their potential in thermotherapy-mediated controlled drug release for cancer therapy. According to this report, the thermotherapeutic properties of MPPs trigger drug release from the drug-MPPs conjugate. Furthermore, the magnetic properties in MPPs also aid in the drug unbinding under an external magnetic field [137].

In addition, the MPPs have exhibited photothermal ability that enables thermotherapeutic destruction of bacteria [24,108]. Popcorn shaped Fe core-Au shell MPPs have shown their antibacterial properties via a study on the multidrug resistant Salmonella DT104 bacterial strain [24]. The experiment achieved approximately 100% bacteria cell death using 2 mL of bacteria cells, at 1.2 × 10^5^ CFU mL^−1^, incubated with 100 µL of particles and exposed to light (670 nm at 1.5 W cm^−2^) for 10 min. In another study, a mixture containing 150 µL of 1.3 × 10^4^ CFU mL^−1^
*E. coli* cells, with 50 µL of MPPs solution, was exposed to light (670 nm, 2.5 W cm^−2^) for 12 min, and showed approximately 90% cell death [108]. In addition, the magnetic properties of MPPs aid the rapid separation of target pathogens in solution.

### 3.3. Imaging

Iron or gold-based NPs have been utilized as imaging contrast agents under alternating magnetic fields, such as in MRI and computed tomography; specifically, aimed at bio-imaging tumors and enabling their associated cancer diagnosis [111,138,139,140,141,142,143,144]. Iron and gold-based MPPs’ properties are used to upgrade the quality of cancer therapy by facilitating real-time diagnosis or image-guided drug delivery and therapy, and provides clear magnetic particle imaging. The high-quality images enable the control of the MPPs’ position under a magnetic environment. Therefore, once the MPPs reach the target position (tumor or cancer cell), the local environment’s structure is imaged and can help in diagnosis. Moreover, in drug delivery, observable drug release improves control of the drug dose and thereby ensures more effective treatment. Therefore, by imaging, a drug can be monitored at the tumor’s location and the drug’s concentration controlled [21,28,51,56,104] (Figure 5). For example, in Tomitaka et al.‘s research, MPPs demonstrated image-guided, near-infrared (NIR) responsive, and triggered drug release capabilities [51]. The MPPs produced clear magnetic particle imaging that enabled the MPPs to be guided to the correct position, and thereby deliver the drug to the required location. The Au parts in MPPs act as drug binders that release the drug under NIR stimulation. Thus, MPPs are candidates for applications in image-guided drug delivery therapy. These novel imaging properties could improve human health care in the future.

## 4. Conclusion and Future Perspectives

The above discussions demonstrate that MPPs are multifunctional nanomaterials that combine the synergistic effects of their magnetic and plasmonic parts. MPPs have improved the quality of existing imaging techniques and have the potential to monitor exceptionally low concentration targets and provide real-time delivery and release of therapeutic drugs. The latter has already been successfully applied at the in vitro and in vivo levels in rats. MPPs can be efficiently used in various systems to provide improved disease diagnosis, monitoring, and treatment. With real-time control, drug dose can be easily monitored, and release rates appropriately adjusted for each individual situation.

Application MPPs in thermotherapy is further widely performed due to not only high effectiveness but also less side effects. For example, in the case of intracranial thermotherapy of glioblastoma multiforme, magnetic NPs combined with external beam radiotherapy did not produce the side effects commonly seen with traditional treatments, such as headaches, nausea, vomitus, and allergic reactions. Additionally, no neurological deficits or infections were evident in the treated regions [112]. This study has shown the potential of next-generation MPPs for cancer treatments that decrease side effects and aid in healing.

MPPs also exhibit potential antibacterial properties. Bacterial inhibition without the use of antibiotics will help prevent the evolution of bacteria capable of defeating antibiotics. Overall, the potential and prospective applications of MPPs could bring significant benefits in the future.

Beyond this, there are many reports that have discussed the application of machine and artificial intelligence in the SERS technique [145,146,147,148]. It has provided a novel idea that it is possible for artificial intelligence which learns using MPPs in bioapplication to raise a new approach in future chemistry.

## Figures and Tables

**Figure 1 nanomaterials-11-01215-f001:**
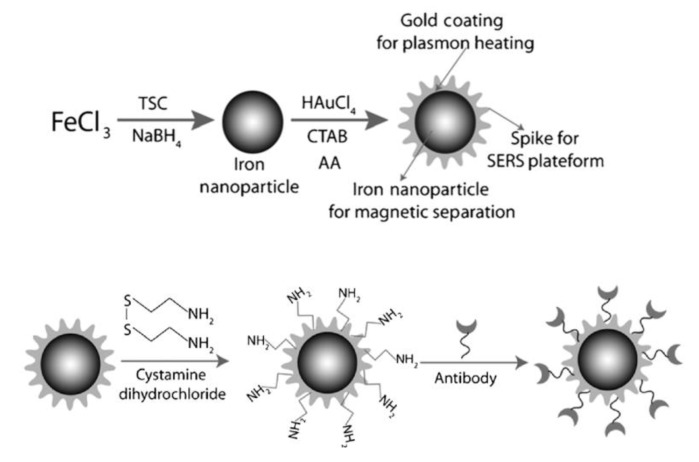
Popcorn shaped, magnetic core, gold shell nanoparticle synthesized and modified with an antibody. Reprinted with permission from ref [24]. Copyright © 2013 WILEY-VCH Verlag GmbH & Co. KGaA, Weinheim.

**Figure 2 nanomaterials-11-01215-f002:**
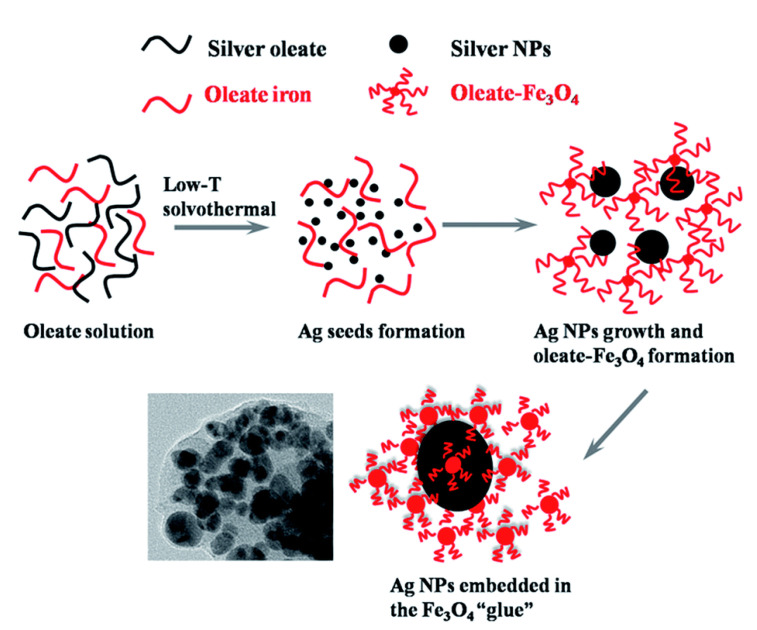
Magnetic Fe3O4 NPs and plasmonic Ag NPs forming an MPPs cluster. Reprinted with permission from ref [50]. Copyright © 2015 The Royal Society of Chemistry.

**Figure 3 nanomaterials-11-01215-f003:**
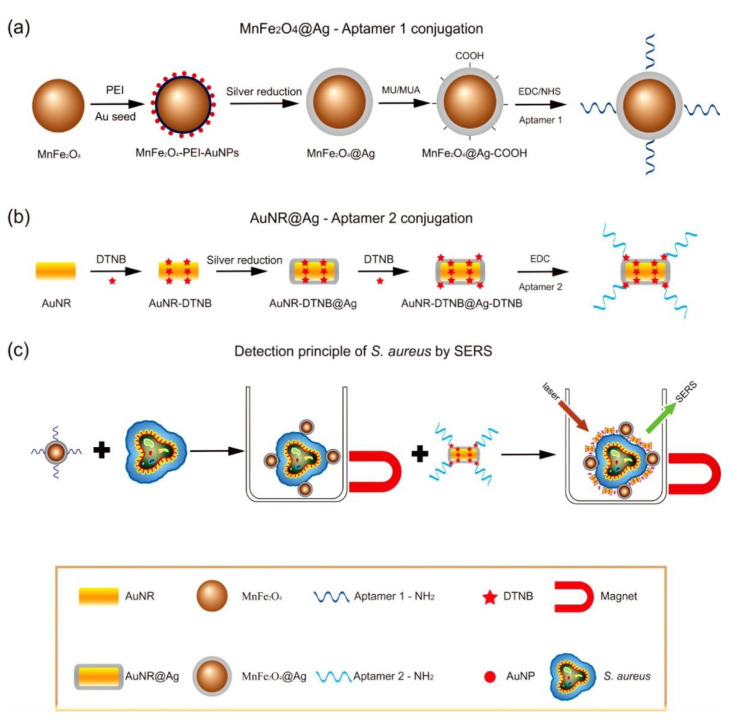
Scheme of Staphylococcus aureus detection by SERS using MPPs. (**a**) Synthesis of monodispersed silver-coated magnetic nanoparticles and their conjugation with aptamer 1; (**b**) Synthesis of gold core/silver shell plasmonic nanoparticles, labeled with SERS tag (DTNB), and their conjugation with aptamer 2; (**c**) operating principle for S. aureus detection. Reprinted with permission from ref [25]. Copyright © 2015 American Chemical Society.

**Figure 4 nanomaterials-11-01215-f004:**
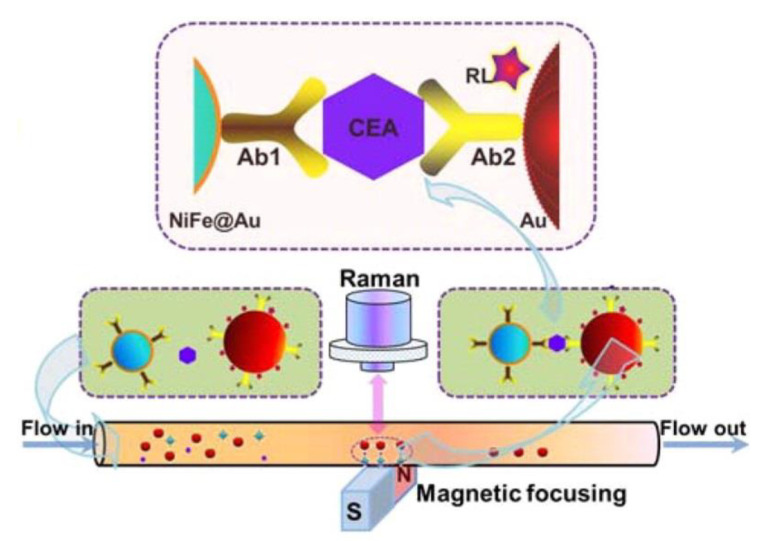
Illustration of SERS detection of cancer biomarker CEA (carcinoembryonic antigen). Using functional nanoprobes consisting of Au coated NiFe magnetic nanoparticles (NiFe@Au), capture antibodies (Ab1), detection antibodies (Ab2), and Raman labels (RL). Reprinted with permission from ref [101]. Copyright © 2015 American Chemical Society.

**Figure 5 nanomaterials-11-01215-f005:**
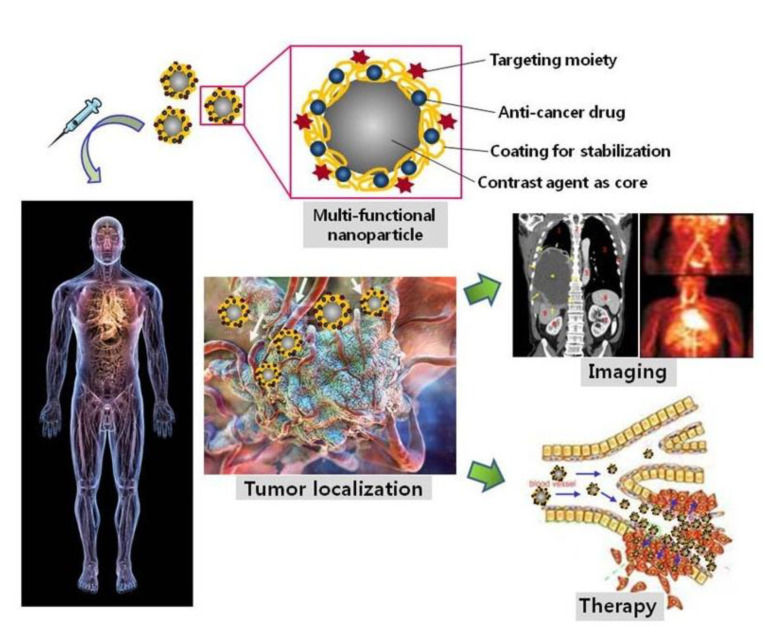
Nanomaterial as a drug carrier and imaging enhancer. Reprinted with permission from ref [56]. Copyright © The Korean Society for Molecular and Cellular Biology.

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
