# Peer review of "Recent Advances in Surface-Enhanced Raman Scattering Magnetic Plasmonic Particles for Bioapplications"

_nanomaterials, 2021, doi:10.3390/nano11051215_

Round 1
Reviewer 1 Report
This manuscript first introduced the technique of surface-enhanced Raman scattering (SERS) and states the shortcomings of magnetic NPs using for bio-applications then bring up the magnetic plasmonic nanoparticles (MPPs). Then it lists a variety of MPPs and gives several examples of designed MPPs and their synthesis processes. The detailed examples of SERS with MPPs are used for bio-applicaitions in medical scenarios, and summarizes that MPPs can be efficiently used in various systems to provide improved disease diagnosis, monitoring, and treatment, have a great application prospect. As a review article, the references of this article are full and accurate, and all content in this article is highly related to the main idea, which can be published on Nanomaterials. However, there are several concerns that may improve the manuscript,
- The MPPs is equivalent to “magnetic plasmonic particles” in title and abstract, but “magnetic plasmonic nanoparticles” in line 55 of Section 1.
- It would be better if there are more analysis of why this type of MPPs is suitable for this medical scenario.
Author Response
Dear Referee,
Please see the attachment as our answer to your comments
Sincerely yours

Reviewer 2 Report
A wide variety of applications of magnetic plasmonic particles(MMP) is discussed here. In this respect this is a very good review paper. However this paper lacks quantitative aspects such as sensitivity, specificity and cost for each application and comparison of various approaches. For example a simple aggregated nano silver particles on a piece of paper can almost do everything MMP does without a need for concentration at extremely low cost using thick film technology. Please check various patents and papers by Som Tyagi, et al.
Method for the formation of SERS substrates
US Patent 8,559,002
and others.
Also recently utilization of machine learning to data obtained from sensors such as SERS is greatly improving the outcome.
Please see
Deep learning and artificial intelligence methods for Raman and surface-enhanced Raman scattering, Felix Lussier, etal.
I think these issues needs to be addressed in this review paper.
Author Response
Dear referee,
Please see the attachment as our answer to your comments
Sincerely yours

Round 2
Reviewer 2 Report
Good job you have answered my concerns.